# Assessing Serum Pepsinogen and *Helicobacter pylori* Tests for Detecting Diffuse-Type Gastric Cancer: Insights from a Large-Scale and Propensity-Score-Matched Study in Republic of Korea

**DOI:** 10.3390/cancers17060955

**Published:** 2025-03-12

**Authors:** Seon Hee Lim, Nayoung Kim, Yonghoon Choi, Ji Min Choi, Yoo Min Han, Min-Sun Kwak, Goh Eun Chung, Ji Yeon Seo, Sung Min Baek, Hyuk Yoon, Young Soo Park, Dong Ho Lee

**Affiliations:** 1Departments of Internal Medicine, Healthcare System Gangnam Center, Seoul National University Hospital, Healthcare Research Institute, Seoul 06236, Republic of Korea; limsh@snuh.org (S.H.L.); 65846@snuh.org (J.M.C.); umminy@naver.com (Y.M.H.); minsunkwak1@gmail.com (M.-S.K.); gohwom@snu.ac.kr (G.E.C.); suji421@gmail.com (J.Y.S.); 2Department of Internal Medicine, Research Center for Sex- and Gender-Specific Medicine, Seoul National University Bundang Hospital, Seongnam 13620, Gyeonggi-do, Republic of Korea; moonsin12345@naver.com (Y.C.); pnu@hanmail.net (S.M.B.); bodnsoul@hanmail.net (H.Y.); dkree@snubh.org (Y.S.P.); dhljohn@yahoo.co.kr (D.H.L.); 3Department of Internal Medicine, Liver Research Institute, Seoul National University College of Medicine, Seoul 03080, Republic of Korea

**Keywords:** pepsinogens, gastric cancer, diffuse type, adenoma, *Helicobacter pylori*

## Abstract

Serum pepsinogen tests were positively associated with gastric cancer (GC) occurrence after adjustment for age, sex, body mass index, *Helicobacter pylori* status, salty food and alcohol intake, smoking, and family history of GC before and after propensity-score matching. Specifically, serum pepsinogen II (PGII) showed a greater association with diffuse-type GC, and the pepsinogen ratio showed a greater association with intestinal-type GC as well as diffuse-type GC. The combination of high PGII and positive *Helicobacter pylori* status was associated with a heightened risk of early-stage diffuse-type GC, especially in young female participants. The strengths of this study include its large sample size, use of propensity-score matching, and adjustment for various confounding factors. This study’s findings may be valuable in developing effective and targeted screening programs, especially in regions with high GC incidence rates.

## 1. Introduction

The incidence of cancer and its associated mortality have increased over the past several decades. Gastric cancer (GC) remains a significant concern worldwide, ranking fifth in incidence and third in mortality among solid malignancies globally [1]. According to a recent meta-analysis, the crude global prevalence of *Helicobacter pylori* (HP) has reduced from 52.6% before 1990 to 43.9% from 2015 to 2022 in adults, and the incidence of GC has decreased globally and in various countries where the prevalence of HP infection has declined [2]. Eastern Asia has the highest GC incidence rate, at 45.7 cases per 100,000 persons, accounting for 60.3% of new GC cases and 56.6% of GC-related deaths worldwide [1]. Although the incidence and mortality of GC have been declining owing to the increased diagnosis of precursor lesions and early GC following the introduction of nationwide screening programs [3,4,5], as well as advancements in treatment techniques, the age-standardized incidence rates of GC in 2019 were still higher than those in other countries, at 28.67, 28.29, and 30.64 per 100,000 individuals in Korea, Japan, and China, respectively [6]. Considering that early diagnosis is key in GC treatment, various efforts are being made to enhance early detection strategies. Population screening has recently been conducted in Korea, Japan, and China [3,4,5]. In Korea, a structured, nationwide GC screening program, including esophagogastroduodenoscopy (EGD) or upper gastrointestinal series designed to detect GC, has been conducted biannually for individuals over 40 years old since 2001 [3]. EGD screening can potentially prevent GC through early diagnosis and treatment; however, conducting it for the entire population is challenging. Mass screening for GC requires a simpler, less invasive, and more cost-effective method. Diverse biomarkers, such as host genetic polymorphism in cytokine genes (e.g., interleukin-1β and interleukin-8 polymorphisms) [7,8] and HP-related factors (e.g., cytotoxin-associated gene A [*cagA*] and vacuolated cytotoxin A) [9], have been used to screen for GC. However, none of these biomarkers have reliably identified individuals with a high risk of GC.

Japan has demonstrated the effectiveness of risk stratification for GC using a serum screening system, including a serum pepsinogen (PG) series with or without an anti-HP antibody (HPIgG) [10,11,12,13], to address the lack of EGD processing capacity. PG levels reflect the morphological and functional status of the gastric mucosa and the extent of atrophic changes in the stomach mucosa [10]. Gastric chief cells and mucous neck cells produce two biochemically distinct PGs, pepsinogen I (PGI) and pepsinogen II (PGII), with the latter also originating from pyloric and Brunner’s gland cells [14,15]. HP colonizes the gastric mucosa, triggering a series of inflammatory responses and significantly increasing GC risk [16,17,18]. Recognizing this long disease progression potentially allows for the early detection of precancerous lesions and timely intervention [13]. Both PGI and PGII levels initially increase. As gastric atrophy develops, the chief cells are replaced by pyloric glands, leading to decreased PGI levels, whereas PGII levels remain relatively unaffected. Consequently, the PGI-to-PGII ratio (PGR) also decreases. Because low PGI and PGR levels reflect gastric atrophy, these markers have been studied to identify high-risk patients with conditions such as atrophic gastritis or intestinal metaplasia [19,20] or to detect neoplastic lesions at an earlier stage [21]. Mechanistically, these markers are expected to be particularly useful for diagnosing intestinal-type gastric cancer (IGC), which follows Correa’s cascade [16,17]. In contrast, evidence on the usefulness of PG in diffuse-type GC (DGC), which may have a stronger genetic component and a weaker association with environmental factors than IGC, has been generally lacking or questionable [11,12,22,23,24,25]. Research on DGC and all types of PGs has continued globally, especially in Japan; however, Some studies have suggested a link between DGC and PGII among all kinds of PG [26,27,28], while another study found that PGII was not an appropriate biomarker for GC screening [29]. Furthermore, few studies have examined the utilization of PG values for DGC diagnosis in the Korean population. DGC comprises a relatively high proportion of total GC (38.3–42.1%) in Korea [30,31] and occurs more commonly in younger people, but they are less likely to undergo EGD because they are less concerned about GC risk. Moreover, some studies have suggested racial differences in the use of PG values for GC diagnosis [30]. Thus, the role of PG values in GC development, especially its utility as a biomarker for DGC, must be redefined to stratify patients who should undergo EGD in South Korea.

Some studies have highlighted the limitations of diagnostic accuracy when using serologic tests alone [32], and others have suggested that assessing the HP infection status concurrently is necessary for clinical utility [33,34], showing a high diagnostic rate of GC in past or current HP infection groups. Recently, our group reported that PGII ≥ 20 ng/mL and a positive HP status may help detect early-stage DGC (DGC-E) in a young female group in a tertiary hospital, although not in the general population [35]. In addition, a recent study has reported the relationship between HP infection and the progression of DGC [36]. Therefore, the aims of this study were (1) to investigate the association of serum PGs, including PGII, with GC occurrence, after adjusting for other GC risk factors, using large-scale health checkup data, and (2) to determine the risk of GC occurrence according to the combination of each PG value and HP status, particularly focusing on DGC.

## 2. Materials and Methods

### 2.1. Study Population

This prospective observational cohort study was conducted at two hospitals, Seoul National University Hospital (SNUH) Gangnam Center (SNUHGC) and Seoul National University Bundang Hospital (SNUBH). A total of 27,974 participants aged 18–93 years who had undergone testing for serum PG levels, HP status, and/or EGD on the same day during a health screening at these two hospitals between May 2003 and February 2022 were initially included. The exclusion criteria were a history of gastrectomy or endoscopic dissection treatment for GC; recent use (1 month prior to enrollment) of proton-pump inhibitors, potassium-competitive acid blockers, non-steroidal anti-inflammatory drugs, or steroids; renal dysfunction; any other upper gastrointestinal neoplasms (e.g., esophageal cancer, neuroendocrine tumor, or gastric lymphoma); and no results for HP status or EGD. The final number of study participants was 23,015. Propensity-score matching (PSM) was performed to control for potential confounders (Figure 1).

### 2.2. Endoscopic Examination and Pathology

One experienced board-certified endoscopist at SNUBH and 15 experienced board-certified endoscopists at SNUHGC performed all health checkup EGDs. The pathology notes were reviewed for patients with GC and gastric adenoma (GA) who underwent surgery or endoscopic removal.

GC was categorized according to the Lauren classification as IGC or DGC [37] and classified based on the depth of invasion of the cancer cells. Early-stage GC was defined as GC that invaded no deeper than the submucosa, irrespective of lymph node metastasis (T1, any N). GA was defined as low- or high-grade dysplasia according to the Vienna classification [38].

### 2.3. Serologic Tests for Helicobacter pylori Antibody and Pepsinogen Series

At SNUBH, serum HPIgG was tested using an enzyme-linked immunosorbent assay (ELISA) (Genedia^®^; Green Cross Medical Science Corp, Yongin, Republic of Korea) during the entire study period. The Genedia^®^ HP ELISA demonstrated 97.8% sensitivity and 92.0% specificity in a Korean population [39]. At SNUHGC, from 2003 to March 2013, HPIgG was measured using an ELISA kit (Radim^®^ Diagnostics, Rome, Italy) and an automatic analyzer, Alisei^®^ (Seac, Pomezia, Italy), which was validated in 2013 in nationwide Korean seroepidemiologic studies [40]. From April 2013 to the present, HPG kits (Immulite^®^ 2000 CMIA, Siemens, UK) were used, which were also validated based on Genedia^®^ in 2017 [41].

Serum levels of PGI and PGII were measured using a latex-enhanced turbidimetric immunoassay (HBi Corp., Seoul, Republic of Korea, imported from Shima Laboratories, Tokyo, Japan) at both hospitals, and PGR and the PGII-to-PGI ratio (PGII/I, reciprocal PGR, PGRr) were calculated. The latex-enhanced turbidimetric immunoassay kit demonstrated that serum PGI and PGR levels were significantly lower when histological atrophic gastritis progressed, with a cut-off value of 3.0 for diagnosing histological atrophic gastritis. A significant correlation was noted between endoscopic and histological atrophic gastritis, with the sensitivity and specificity of endoscopic diagnosis being 65.9% and 58.0% for the antrum and 71.3% and 53.7% for the corpus, respectively [42]. The sensitivity and specificity of a PGR of ≤3.0 for detecting dysplasia or cancer were 55.8–62.3% and 61%, respectively [30]. In addition, this kit was also validated in previous studies [43,44,45].

### 2.4. Helicobacter pylori Status Assessment

In addition to the HPIgG test, HP infection was diagnosed using EGD with biopsies for histology, HP polymerase chain reactions, rapid urease tests, or [^13^C]-urea breath tests, as required. A positive result for any one of these four tests was considered to indicate a current HP infection. Our group reported that the rapid urease test had 96.7% specificity and 80.4% sensitivity [46], and the urea breath test’s sensitivity and specificity for a cut-off value of 2.5‰ were 99.3% and 47.1%, respectively [47].

The HPIgG test was performed for qualitative estimation, particularly when the other four HP tests were negative, and all participants were assessed for HP eradication history. A positive HPIgG result or HP eradication history indicated a past HP infection. Overall, individuals with both current and past HP infection statuses were considered “HP status-positive” throughout their lifetime. Since the so-called “point of no return” beyond which eradication may not prevent further progression of premalignant conditions remains currently undefined [48,49], we decided to include all participants with a history of past HP infection, not just actual infection.

### 2.5. Risk Factor Measurements

Information on medical history, family history (at least one first-degree relative with GC), anthropometric assessment, current or past smoking status (at least one cigarette per day during the previous 12 months), alcohol consumption (≥140 g/week during previous 12 months), and salty food intake habits was obtained through a questionnaire administered to each participant. Participants were categorized into two groups according to smoking status (never-smokers and ever-smokers) and alcohol intake (current habitual use or not). Regarding salty food intake habits, participants were categorized into two groups (“no/mild” [score 0–2] and “severe” [score 3–4]) according to scores based on two questions: “Do you add extra salt or soy sauce to your food when you eat?” and “Do you eat salty foods such as salted seafood, pickled vegetables, and soup-based meals?”. Both questions had three types of answers, including never (score 0), sometimes (score 1), or frequently (score 2).

### 2.6. Statistical Analyses

To compare the baseline characteristics of each group, categorical variables were compared using the chi-square test, Fisher’s exact test, or the Kruskal–Wallis test with Bonferroni correction, and the results are presented as numbers and percentages. Continuous data were compared using a *t*-test or a one-way analysis of variance and are expressed as medians (interquartile ranges). The area under the curve (AUC) and receiver operating characteristic curves were computed to determine the optimal cut-off values of PGs for GC detection. Additionally, to compare diagnostic accuracy, specifically in detecting PGs in subgroups of GC such as IGC and DGC, an AUC ≥ 0.7 and sensitivity and specificity ≥ 70% were regarded as significant.

Given that the occurrence of HP infection varies significantly with age and year of birth, we used PSM with a *k*-nearest neighbors approach to match individuals based on their HP status. Age and year of birth were used as covariates due to their significant clinical impact on HP occurrence. We performed 1:1 matching and calculated the caliper using the standard deviation of logit propensity scores. We performed PSM to match the case and control groups based on their propensity scores, ensuring balanced covariates and enhancing the validity of our causal inferences. This matching approach allowed us to better predict GC and GA by controlling for these confounding variables. The standardized mean differences for age and year of birth changed from 0.22 to 0.01 and from 0.30 to 0.08, respectively, before and after PSM (Appendix A).

To validate the effectiveness of PGs in predicting GC risk, participants were divided into two categories (positive vs. negative) based on the cut-off values of three types (PGIR, PGR, and PGII) from previous studies [10,11,12,30,35,50]. Cut-off values of PGI ≤ 70 ng/mL and PGR ≤ 3 have been widely accepted as indicating an atrophy-positive status and are frequently used for GC screening purposes [10,11,12,50]. Therefore, we classified participants as PGIR-positive when they met the criteria mentioned above and PGIR-negative when they did not. A previous study from Korea reported that a PGI value of ≤ 70 ng/mL had adequate sensitivity (72.4%) but low specificity (20.2%), whereas the sensitivity and specificity of PGR ≤ 3 were 59.2–61.7% and 61.0%, respectively, for GC or GA [30]. Therefore, we used PGR ≤ 3 and categorized participants as PGR-positive if they met the criterion and PGR-negative if they did not. Another Korean study revealed that PGII ≥ 20 ng/mL (an AUC of 0.593 for DGC) was strongly related to DGC-E, particularly in young adult females [35]. In the present study, the ROC curve identified 21 ng/mL (an AUC of 0.706 for DGC) as the optimal PGII cut-off value for diagnosing DGC-E, achieving 66.7% sensitivity and 79.2% specificity. Therefore, we used a cut-off value of PGII ≥ 21 ng/mL to classify participants as PGII-positive. Each of these three types of PG values was then divided into two categories, and odds ratios (ORs) with 95% confidence intervals were calculated using multivariable logistic regression analysis. Statistical analyses were performed using SPSS Statistics version 27.0 (IBM Corp., Armonk, NY, USA) and SAS version 9.4 (SAS Institute, Cary, NC, USA). Statistical significance was set at a *p*-value of <0.05.

## 3. Results

### 3.1. Baseline Characteristics

Among the 23,015 study participants, 1283 and 412 were diagnosed with GC and GA, respectively. GC was categorized as intestinal-type (*n* = 760; 59.2%) or diffuse-type (*n* = 490; 38.2%), whereas 33 participants (2.6%) were classified as having mixed- or unclassified-type GC owing to difficulty in categorization before PSM (Figure 1, Table 1 and Appendix A). In total, 571 participants had early-stage IGC (75.1% of IGC), whereas 239 had DGC-E (48.8% of DGC; *p* < 0.001). The IGC and DGC subgroups did not differ in terms of GC location (cardia vs. non-cardia). Regarding GC treatment, surgical therapy was markedly more common in the DGC subgroup than in the IGC subgroup (77.6% vs. 52.7%), and endoscopic therapy was relatively more common in the IGC subgroup than in the DGC subgroup (40.3% vs. 6.5%; Appendix A). After PSM (Table 2 and Appendix A), 945 participants were categorized into the GC group, 551 into the IGC subgroup (58.3%), 369 into the DGC subgroup (39%), and 285 into the GA group, whereas 14,724 participants formed the control group.

Before PSM (Table 1 and Appendix A), the GA, GC, IGC, and DGC groups were predominantly male compared to the controls; however, the male predominance disappeared in the GA and DGC groups after PSM (Table 2 and Appendix A). The median age of the control group was significantly lower than those of the GA and GC groups, as well as the IGC subgroup, both before and after PSM. A pooled one-way analysis of variance revealed significant differences in smoking and alcohol history as well as family history of GC in the GC groups, especially in the IGC subgroup, before and after PSM. The GA, GC, and two GC subgroups significantly differed from the control group in terms of salty diet and HP status before and after PSM.

Regarding PG series results, the median serum level of PGII in the GC group was higher than those in the GA and control groups before and after PSM (Appendix A). Furthermore, the median PGII level was higher in the DGC subgroup than in the IGC subgroup, which was similar to the median PGI level. The median value of PGR in the GC group was significantly lower than that in the GA and control groups.

### 3.2. Association of Each Pepsinogen Value or Helicobacter pylori Status with Gastric Neoplasm After Propensity-Score Matching

The participants’ PG values were classified into two categories using specific cut-off values. After PSM, their correlation with the GA and GC subtypes was examined. Positive associations were observed in the GA, GC, IGC, and DGC groups with PGII ≥ 21 ng/mL, PGR ≤ 3, and PGIR-positive statuses. Additionally, a positive HP status was associated with an increased risk of GA, GC, IGC, or DGC (Appendix A).

### 3.3. Multivariable-Adjusted Logistic Regression Analysis for Gastric Neoplasms

The three types of PG values were classified into two categories, based on cut-off values, and their correlations with the GA and GC groups, as well as GC subtypes, were examined after adjusting for sex, age, body mass index (BMI), smoking, alcohol intake, salty food intake, family history of GC, and HP status (Table 3).

Before PSM, the association between each gastric neoplastic group and each PG value (PGII-positive, PGR-positive, or PGIR-positive) remained significant after adjusting for the other confounding factors listed above, except the association between PGII-positive statuses and GA, which was not significant (Table 3, top panel). However, after PSM, there was no significant association between PGIR-positive status and all gastric neoplastic groups, whereas the association between PGII-positive status and the DGC-E subgroup and the association between PGR-positive status and the IGC or DGC subgroups remained significant (Table 3, bottom panel). After PSM, the adjusted OR (aOR) of PGII-positive statuses for DGC was 2.31 (*p* < 0.001) and increased to 4.20 for DGC-E (*p* < 0.001; Appendix A). After PSM, the aOR of PGR was 2.88 for IGC and 2.43 for early-stage IGC (Appendix A, bottom panel).

### 3.4. Detection Power of Pepsinogen Values for Diagnosis of Gastric Cancer According to Subtypes

We assessed the detection power of two PG values, PGII and PGR, which exhibited an association with GC after adjustments for various risk factors and PSM. Figure 2A–F show the AUC of PGII for the diagnosis of GC and its subgroups. The highest AUC was observed in females under 40 years with DGC-E (AUC: 0.843; sensitivity: 0.813; specificity: 0.897; Figure 2F).

Given that the AUC of GC occurrence for PGR was <0.5, we chose to use PGRr (PGII/I). The AUC of GC occurrence for PGRr was 0.75 (95% confidence interval: 0.735–0.764), and the optimal PGRr cut-off value was 0.26 in our model, with a sensitivity of 0.67 and a specificity of 0.72. We set the cut-off level of PGRr at 0.26. Figure 2G–L show the AUC of PGRr for the diagnosis of GC and its subgroups. The AUC was greatest in males under 50 years with IGC-A (AUC: 0.866; sensitivity: 0.999; specificity: 0.62; Figure 2L).

We also calculated the positive predictive value (PPV) of PGII-positive and PGR-positive statuses for the diagnosis of GC and GC subtypes; the PPV for the diagnosis of GC was 11.4% for PGII-positive and 15.1% for PGR-positive cases (Appendix A).

### 3.5. Risk Stratification by Combining Pepsinogen Values and Helicobacter pylori Status in Gastric Cancer and Its Subtypes

GC risk was assessed based on four categories combining HP status and each of the two PG values (PGII and PGR), and it showed significance after adjusting for various risk factors for GC. After PSM, the OR for PGII-positive or PGR-positive statuses differed significantly between the categories, with stepwise increments from negative to positive HP status and from negative to positive status for each PG value, after adjustment for other confounding GC risk factors, such as sex, age, BMI, family history of GC, smoking history, alcohol intake, and salt intake. GC risk showed the highest aOR with HP-positive and PGII-positive statuses or HP-positive and PGR-positive statuses (aOR = 6.93 or aOR = 9.66, respectively; all *p* < 0.001; Table 4).

In addition, multivariable analysis was performed according to sex and age group to assess the risk stratification for each GC subtype using a combination of HP statuses and each PG value (Appendix A). Interestingly, the highest risk of incidental DGC-E occurrence was found in the female group under 40 years with both HP-positive and PGII-positive statuses (OR= 25.8; *p* < 0.001; Appendix A), as well as with HP-positive and PGR-positive statuses (OR= 8.6; *p* < 0.001; Appendix A). For the risk of IGC, the OR was higher in males under 50 years with HP-positive statuses and each PG-positive status (OR= 5.18; *p* < 0.001 for PGII [Appendix A] and OR= 10.44; *p* < 0.001 for PGR [Appendix A]).

## 4. Discussion

This large-scale (n = 23,015) case–control study was conducted to validate the utility of PG in diagnosing GC during health checkups. We found that PGII-positive or PGR-positive statuses, but not PGIR-positive statuses, were independent predictors of GC in the general population after PSM. This relationship remained significant even after adjusting for other confounding factors, including HP status. In particular, a PGII-positive status was significantly related to DGC-E, whereas a PGR-positive status was associated with IGC as well as DGC-E.

Our previous study [35], which focused on a cohort from a tertiary hospital, revealed that higher serum PGII levels and positive HP status indicated an increased risk of DGC-E, particularly in young adult females. The present results corroborate these findings, clearly demonstrating that a PGII-positive status was significantly associated with DGC-E, especially in young females, whereas a PGR-positive status was associated with IGC as well as DGC-E after general health screeners were included and PSM was applied.

The findings are generally consistent with previous results, although a few studies have reported conflicting results. Kikuchi et al. [26] concluded that PGII and PGR could be markers for not only IGC but also DGC in a young population (108 GC with 216 age- and sex-matched controls). Yanaoka et al. [27] reported that GC risk increased with a reduced PGI level or a reduced PGR, and the risk of DGC increased as the PGII level increased. Yoshida et al. [51] also identified a significantly elevated DGC risk in a subgroup with high PGII from an HP-infected and atrophy-negative group. However, both studies [27,51] were conducted on the same cohort of middle-aged men, with no adjustments made for other GC risk factors when analyzing the risk stratification for GC and its subgroups. Abnet et al. [52] reported that a lower PGR value is associated with a higher risk of GC but did not analyze the association of PG values and GC risk based on histologic type.

In contrast, Oishi et al. [12] reported a stepwise increase in the adjusted hazard ratio for IGC among four categories according to a combination of PGIR and HP statuses; however, such an association was not found for DGC, and they did not mention a relationship between PGII and GC risk. Parsonnet et al. [18] reported that low PGI or low PGR levels were linked to the development of distal GC, but PGII levels were not linked to GC. However, they could not analyze the association between PG values and DGC development because of an insufficient sample size (26 DGC cases among 136 GC cases). The Japan–Hawaii Cancer Study Group found that low PGI levels, with or without low PGR, increased the risk of IGC in males, but there was no difference in males with DGC compared to control individuals in terms of PGI or PGR. Moreover, they did not mention an association between PGII and GC [22].

Another notable finding was that the aOR value for GC tended to increase progressively from statuses of HP-negative to HP-positive and from PG-negative to PG-positive depending on the combination of HP status and each of the two PG values, after adjusting for sex, age, BMI, family history of GC, smoking history, alcohol intake, and salt intake, and after PSM. Before adjusting for other confounding factors, the highest OR value was observed in the combination of HP-negative and PGR-positive statuses, as shown in other studies [24,51,53]; however, following adjustment for various confounding factors, the aOR value was the highest for groups with both HP-positive and PGII-positive or PGR-positive statuses. This may be a product of ensuring balanced covariates and enhancing the validity of our causal inferences, which allowed us to better predict GC risk. Parsonnet et al. [18] reported that a combination of HP-positive and atrophy-positive statuses showed the highest OR for GC, similar to our findings. Our identification of biomarker combinations with the greatest risk differed from those reported in most Japanese studies [24,51,53], likely due to differences in exclusion criteria. In most studies that have investigated GC risk by combining PG and HP status, HP-eradicated subjects were excluded. However, in this study and the study by Parsonnet et al. [18], HP-eradicated subjects were not excluded. Therefore, those with actual HP infection as well as those with prior infection were classified as HP-positive in the current study.

Moreover, we performed a risk stratification analysis using four categories based on combinations of HP statuses and PG values to elucidate their influence on GC risk through the interaction of HP status and atrophy or inflammation level and to identify the most effective model for predicting each subtype of GC. The OR for DGC-E was the highest for HP-positive statuses and each PG value-positive status in individuals under 40 years of age, particularly in the female group, whereas the OR for IGC was the highest in HP-positive and PG value-positive individuals under 50 years of age, especially in the male group.

We also found that the PPV of each PG value for detecting GC ranged from 11.4% to 15.1%, which was slightly higher than that reported in other studies [54,55]. Mizuno et al. [54] used cut-off levels of PGI ≤ 30 ng/mL and PGR ≤ 2.0 and reported a PPV of 1.4%, which was similar (1.8%) to that reported for the direct X-ray method; when cut-off levels were changed to ≤70 ng/mL and PGR ≤ 3.0, respectively—the same as “PGIR-positive status” in the present study—the researchers reported a PPV of 0.7%. Tong et al. [55] reported that the PPV of PGI ≤ 43.5 ng/mL was 1.5% and that of PGR ≤ 4.7 for GC was 2.7%. Globally, low PPVs have been observed in the studies on PG and GC.

In the present study, the sex ratio (male/female) was 1:0.8 in the DGC subgroup and 1:2 in the DGC subgroup under 40 years of age, compared to 3:1 in IGC and 2:1 for all GC cases. In addition, there was a relatively higher proportion of HP-positive (90%) cases in the DGC subgroup than in the IGC subgroup (83%; *p* < 0.001, Table 1). Our finding that DGC more frequently affects young females and has a higher HP-positive status is similar to that reported in another health checkup study in Korea [31]. One possible reason for the high incidence of DGC in young females may be its association with female sex hormones such as estrogen; the higher proportion of HP-positive status in DGC may be associated with active inflammation caused by HP, which can increase PGII levels. Recently, a mechanism by which HP and estrogen act simultaneously to promote DGC progression has been identified, which may explain the higher incidence of DGC in premenopausal women with physiologically elevated hormone levels [36]. Kang et al. [36] found that estrogen levels increased the expression of an oncogene in estrogen receptor α-positive DGC and that *cagA* toxin-secreting HP enhanced the effects of estrogen in DGC.

Our findings may have useful clinical implications. Serologic tests, which are relatively simple, easy, and economical, should be performed for PG series and HP status (HPIgG) in the general population under 40 years of age, as they are not included in national cancer screenings in Korea, followed by EGD screening for those who are PGII-positive, PGR-positive, or HP-positive. Simulating such a stepwise approach with our data, 57 (90.5%) of 63 patients with GC under 40 years old could be diagnosed, and 57 (6.7%) of 857 subjects who underwent screening for EGD would be diagnosed with GC. A total of six GC cases were missed due to failure to filter out by serologic results under 40 years (Appendix A), and the false negative rate (1-sensitivity) was 9.52%. In other words, if an endoscopy is not performed solely based on serologic results, approximately 9.5% of GC patients may be missed. This is an area that needs to be studied with caution. To compensate for this, the presence or absence of symptoms and other GC risk factors such as a family history of GC or smoking should be sufficiently considered.

The present study has several strengths. Its large sample population, exceeding 23,000 individuals, including over 1200 GC cases, enabled a comprehensive analysis across group categories and histological subtypes while accounting for potential confounding effects. Moreover, the inclusion of health checkup participants mitigates the case selection bias encountered in clinical series from hospital cases. Additionally, we performed PSM to match the case and control groups based on their propensity scores and enable better prediction of GC and GA by controlling for these confounding variables. Finally, since the GC stages were classified based on histology and HP status, the role of PGs in predicting GC risk was evaluated in detail.

Despite these advantages, this study has a few limitations. First, a positive HP status did not exclusively represent active infections. In this study, tests reflecting active HP infection, such as the rapid urease test, urea breath test, culture, HP polymerase chain reaction, or histology, were not performed on all participants. In cases where results from such active tests were unavailable, the serological test results (HPIgG) were used. Those with a history of HP eradication were also included. It was recognized that a positive HPIgG result or a history of HP eradication indicates a past HP infection and that a past infection could develop into a precancerous lesion. Second, this was a case–control study, although the participants were enrolled prospectively; therefore, the observational design did not include follow-up data, and selection bias was unavoidable. Nevertheless, the relatively large sample size of the present study may have compensated for the validity of our conclusions. In the future, a well-designed prospective cohort study, preferably a randomized control trial, is required to validate our results. Third, *cagA* toxin-producing HP, which is probably related to DGC risk, was not investigated in the present study. However, our group has previously found that HP infection in South Koreans is closely related to highly virulent strains; most HP colonies include *cagA*-positive strains (87.2%) and vacuolating cytotoxin genes (92.9–100%) [56].

## 5. Conclusions

PG values (PGII or PGR) and HP status could be a screening tool for identifying high-risk individuals for GC. In particular, the combination of high PGII levels and positive HP status was related to a heightened risk of DGC-E, especially in the young female group. Low PGR-positive and positive HP statuses were associated with IGC as well as DGC-E.

Given the global significance of GC as a major cause of cancer-related mortality, our study findings may have implications for designing more effective and targeted screening programs, especially in regions with high GC incidence rates.

## Figures and Tables

**Figure 1 cancers-17-00955-f001:**
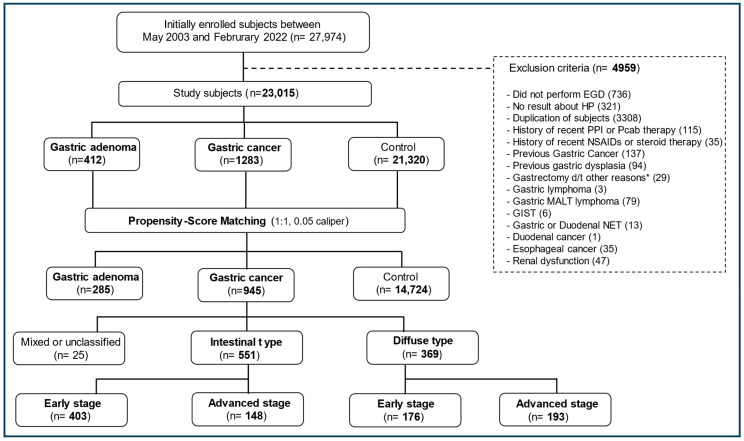
Flow diagram of the study. * Gastrectomy due to complications (bleeding, perforation, etc.) of benign gastric ulcer. EGD, esophagoduodenoscopy; HP, *Helicobacter pylori*; PPI, proton-pump inhibitor; Pcab, potassium-competitive acid blocker; NSAIDs, non-steroidal anti-inflammatory drugs; d/t, due to; MALT, mucosa-associated lymphoid tissue; GISTs, gastrointestinal stromal tumors; NETs, neuroendocrine tumors.

**Figure 2 cancers-17-00955-f002:**
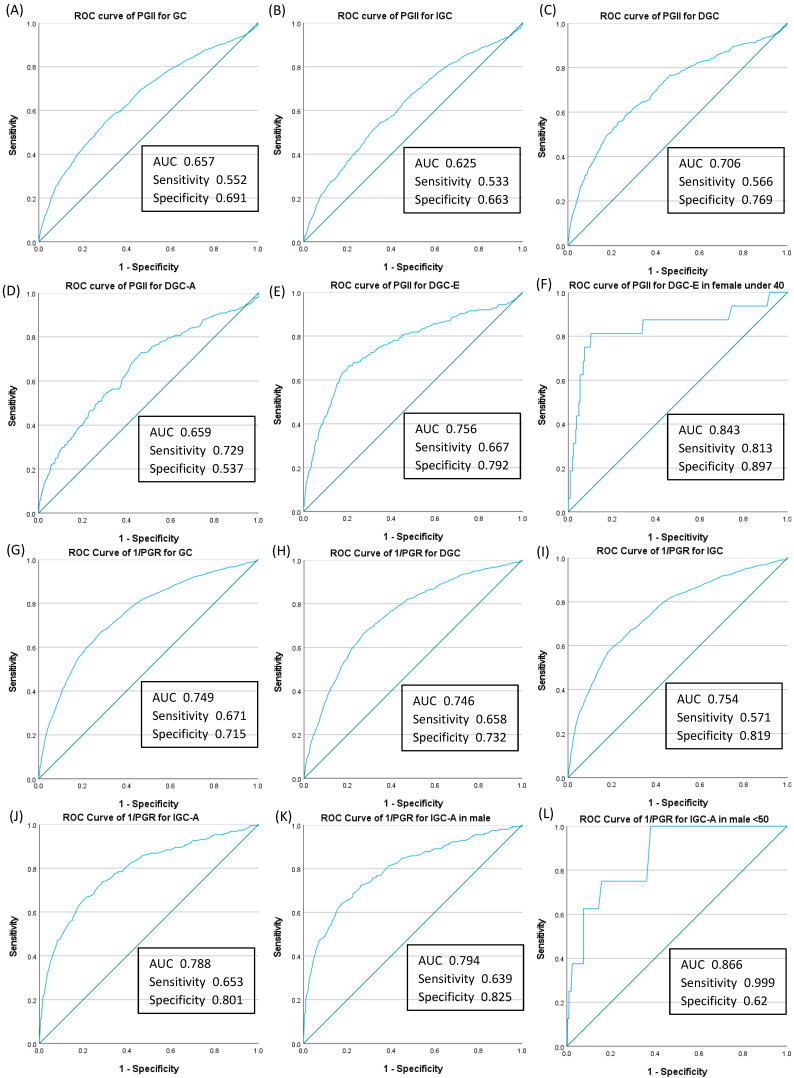
Receiver operating characteristic curve and corresponding area under the curve of each pepsinogen value for the diagnosis of gastric cancer. The AUC for serum pepsinogen II (PGII) showed significant sensitivity and specificity for six cases: (**A**) total GC, (**B**) IGC, (**C**) DGC, (**D**) DGC-A, (**E**) DGC-E, and (**F**) DGC-E in females under 40 years of age. The greater the progress in alphabetical order, the higher the diagnostic power (**A**–**F**). The AUC for reciprocal pepsinogen ratio (1/PGR) showed significant sensitivity and specificity for six cases: (**G**) total GC, (**H**) DGC, (**I**) IGC, (**J**) IGC-A, (**K**) IGC-A in males, and (**L**) IGC-A in males under 50 years. The greater the progress in alphabetical order, the higher the diagnostic power (**G**–**L**). ROC, receiver operating characteristic; AUC, area under the curve; PG, serum pepsinogen; PGII, serum pepsinogen II ≥ 21 ng/mL; GC, gastric cancer; IGC, intestinal-type gastric cancer; DGC, diffuse-type gastric cancer; DGC-A, diffuse-type gastric cancer, advanced stage; DGC-E, diffuse-type gastric cancer, early stage; IGC-A, intestinal-type gastric cancer, advanced stage; 1/PGR, reciprocal pepsinogen ratio.

**Table 1 cancers-17-00955-t001:** Baseline characteristics of the subjects before propensity-score matching.

Characteristics	Control	GC	*p*-Value	Intestinal-Type GC	*p*-Value	Diffuse-Type GC	*p*-Value	*p*- Value *
Number (%)	21,320 (92.6)	1283 (5.6)		760 (3.3)		490 (2.1)		
Sex			**<0.001**		**<0.001**		**0.012**	**<0.001**
Female	8336 (39.1)	404 (31.5)		176 (23.2)		219 (44.7)		
Male	12,984 (60.9)	879 (68.5)		584 (76.8)		271 (55.3)		
Age, years	54 [13.0]	63 [17.0]	**0.001**	65.0 [13.0]	**0.001**	55.0 [19.0]	0.229	**0.001**
BMI (kg/m^2^)	23.7 [4.0]	23.3 [4.2]	**0.005**	23.6 [4.2]	0.857	22.7 [4.2]	**0.001**	**0.001**
Smoking			**<0.001**		**<0.001**		**0.008**	**<0.001**
Never	10,638 (49.9)	478 (37.3)		250 (32.9)		217 (44.3)		
Ever	10,437 (49.0)	801 (62.4)		507 (66.7)		272 (55.5)		
Alcohol			**<0.001**		**<0.001**		0.183	**<0.001**
Never	8067 (37.8)	399 (31.1)		219 (28.8)		171 (34.9)		
Ever	13,163 (61.7)	876 (68.3)		536 (70.5)		317 (64.7)		
Salty diet			**<0.001**		**<0.001**		**<0.001**	**<0.001**
No/mild	19,052 (89.4)	595 (46.4)		237 (31.2)		336 (68.6)		
Strong	1909 (8.8)	687 (53.5)		523 (68.8)		153 (31.2)		
Family history of GC			**<0.001**		**<0.001**		**0.019**	**<0.001**
Negative	18,233 (85.5)	1009 (78.6)		582 (76.6)		401 (81.8)		
Positive	3068 (14.4)	273 (21.3)		177 (23.3)		89 (18.2)		
HP status			**<0.001**		**<0.001**		**<0.001**	**<0.001**
Negative	7811 (36.6)	190 (14.8)		129 (17.0)		52 (10.6)		
Positive	13,509 (63.4)	1093 (85.2)		631 (83.0)		438 (89.4)		
PG series								
PGI, ng/mL	52.3 [27.5]	46.7 [49.4]	0.331	40.0 [41.3]	**0.001**	55.9 [56.5]	**0.001**	**0.001**
PGII, ng/mL	10.8 [10.1]	14.1 [16.0]	**0.001**	12.6 [13.7]	**0.001**	18.4 [18.0]	**0.001**	**0.001**
PGI/II ratio	4.8 [2.5]	3.1 [2.5]	**<0.001**	2.9 [2.7]	**0.001**	3.2 [2.1]	**0.001**	0.261

n (percentage); median [interquartile range]; GC, gastric cancer; HP, *Helicobacter pylori*; PG, pepsinogen; PGI, serum pepsinogen I; PGII, serum pepsinogen II. The basal characteristics of the subjects with gastric adenoma are shown in Appendix A, and subjects with missing data are shown in Appendix A. Values that are statistically significant, with a *p*-value less than 0.05, are displayed in bold. *p*-value * refers to the statistical test between intestinal-type gastric cancer and diffuse-type gastric cancer and was corrected with Bonferroni correction for multiple testing. Salty diet scores: “no/mild” (score 0~2) and “severe” (score 3~4) assessed via two questions [“Do you add extra salt or soy sauce to your food when you eat?” and “Do you eat salty foods such as salted seafood, pickled vegetables, and soup-based meals?”] which had three kinds of answers [“never (score 0), sometimes (score 1), frequently (score 2)”], in the questionnaire.

**Table 2 cancers-17-00955-t002:** Baseline characteristics of the subjects after propensity-score matching.

Characteristics	Control	GC	*p*-Value	Intestinal-Type GC	*p*-Value	Diffuse-Type GC	*p*-Value	*p*- Value *
Number (%)	14,724 (92.3)	945 (5.9)		551 (3.5)		369 (2.3)		
Sex			**0.001**		**<0.001**		0.063	**<0.001**
Female	5816 (39.5)	308 (32.6)		131 (23.8)		172 (46.6)		
Male	8908 (60.5)	637 (67.4)		420 (76.2)		197 (53.4)		
Age, years	52 [14]	60 [18]	**<0.001**	64 [15]	**<0.001**	52 [18]	0.217	**<0.001**
BMI (kg/m^2^)	23.6 [4.0]	23.2 [4.1]	**<0.001**	23.4 [3.7]	0.055	22.7 [4.6]	**<0.001**	**0.003**
Smoking			**<0.001**		**<0.001**		0.156	**0.004**
Never	7355 (50.0)	353 (37.4)		185 (33.6)		162 (43.9)		
Ever	7204 (48.9)	590 (62.4)		364 (66.1)		207 (56.1)		
Alcohol			**0.008**		**0.014**		0.871	0.226
Never	5485 (37.3)	290 (30.7)		161 (29.2)		123 (33.3)		
Ever	9170 (62.3)	648 (68.6)		385 (69.9)		245 (66.4)		
Salty diet			**<0.001**		**<0.001**		**<0.001**	**<0.001**
No/mild	13,218 (89.8)	452 (47.8)		181 (32.8)		252 (68.3)		
Severe	1243 (8.4)	492 (52.1)		370 (67.2)		116 (31.4)		
Family history of GC			**<0.001**		**<0.001**		0.452	0.222
Negative	12,637 (85.8)	746 (78.9)		424 (77.0)		300 (81.3)		
Positive	2072 (14.1)	198 (21.0)		126 (22.9)		69 (18.7)		
HP status			**<0.001**		**<0.001**		**<0.001**	**0.002**
Negative	7715 (52.4)	206 (21.8)		136 (24.7)		60 (16.3)		
Positive	7009(47.6)	739(78.2)		415(75.3)		309 (83.7)		
PG series								
PGI, ng/mL	58.2 [25.8]	52.1 [48.5]	0.666	44.7 [41.2]	**0.018**	61.6 [53.9]	**0.004**	**<0.001**
PGII, ng/mL	10.0 [8.5]	17.4 [19.6]	**<0.001**	15.4 [17.0]	**<0.001**	20.7 [20.5]	**<0.001**	**0.001**
PGI/II ratio	5.1 [2.4]	3.0 [2.1]	**<0.001**	2.9 [2.4]	**<0.001**	3.1 [2.0]	**<0.001**	0.755

n (percentage); median [interquartile range]; GC, gastric cancer; HP, *Helicobacter pylori*; PG, pepsinogen; PGI, serum pepsinogen I; PGII, serum pepsinogen II. The basal characteristics of the subjects with gastric adenoma are shown in Appendix A, and subjects with missing data are shown in Appendix A. Values that are statistically significant, with a *p*-value less than 0.05, are displayed in bold. *p*-value * refers to the statistical test between intestinal-type gastric cancer and diffuse-type gastric cancer and was corrected with Bonferroni correction for multiple testing. Salty diet scores: “no/mild” (score 0~2) and “severe” (score 3~4) assessed via two questions [“Do you add extra salt or soy sauce to your food when you eat?” and “Do you eat salty foods such as salted seafood, pickled vegetables, and soup-based meals?”] which had three kinds of answers [“never (score 0), sometimes (score 1), frequently (score 2)”], in the questionnaire.

**Table 3 cancers-17-00955-t003:** Multivariable-adjusted logistic regression analysis for gastric neoplasms according to pepsinogen values before and after propensity-score matching.

**Before**	**PGII, aOR**			**PGIR, aOR**			**PGR, aOR**		
**PSM**	neg	pos	*p*-Value	neg	pos	*p*-Value	neg	pos	*p*-Value
GA	1	0.98 (0.77–1.24)	0.873	1	4.79 (3.87–5.93)	**<0.001**	1	4.75 (3.82–5.92)	**<0.001**
GC	1	2.17 (1.89–2.48)	**<0.001**	1	2.97 (2.59–3.04)	**<0.001**	1	3.69 (3.22–4.22)	**<0.001**
IGC	1	1.53 (1.27–1.84)	**<0.001**	1	3.47 (2.89–4.16)	**<0.001**	1	3.85 (3.22–4.62)	**<0.001**
DGC	1	3.01 (2.49–3.66)	**<0.001**	1	2.57 (2.10–3.14)	**<0.001**	1	3.59 (2.95–4.36)	**<0.001**
**After**	**PGII, aOR**			**PGIR, aOR**			**PGR, aOR**		
**PSM**	neg	pos	*p*-Value	neg	pos	*p*-Value	neg	pos	*p*-Value
GA	1	0.70 (0.50–0.97)	**0.034**	1	1.47 (0.89–2.43)	0.134	1	3.91 (2.28–6.73)	**<0.001**
GC	1	1.66 (1.38–2.00)	**<0.001**	1	1.32 (0.99–1.75)	0.055	1	2.50 (1.86–3.34)	**<0.001**
IGC	1	1.12 (0.86–1.46)	0.401	1	1.34 (0.89–2.01)	0.159	1	2.88 (1.89–4.40)	**<0.001**
DGC	1	2.31 (1.78–2.98)	**<0.001**	1	1.38 (0.96–1.98)	0.083	1	1.96 (1.34–2.85)	**<0.001**

Multivariables adjusted for sex, age, body mass index, family history of gastric cancer, salty food, smoking, alcohol intake, and *Helicobacter pylori* status. Values that are statistically significant, with a *p*-value of < 0.05, are displayed in bold. PGII neg (PGII- negative), serum pepsinogen II < 21 ng/mL; PGII pos (PGII-positive), serum pepsinogen II ≥ 21 ng/mL; PGIR neg (PGIR-negative), serum pepsinogen I > 70 or pepsinogen ratio > 3; PGIR pos (PGIR-positive), serum pepsinogen I ≤ 70 and pepsinogen ratio ≤ 3; PGR neg (PGR-negative), pepsinogen ratio > 3; PGR pos (PGR-positive), pepsinogen ratio ≤ 3; aOR, adjusted odds ratio; GA, gastric adenoma; GC gastric cancer; IGC, intestinal-type gastric cancer; DGC, diffuse-type gastric cancer.

**Table 4 cancers-17-00955-t004:** Effects of combination of *H. pylori* status and each pepsinogen value on risk of gastric cancer after propensity-score matching.

Combining Condition	n/N	Incidence of GC (%)	OR (95% CI)	*p*-Value	aOR (95% CI)	*p*-Value *
HP (−)/PGII (−)	172/7502	2.24	1		1	
HP (+)/PGII (−)	391/5040	7.2	3.38 (2.82–4.06)	**<0.001**	3.22 (2.65–3.91)	**<0.001**
HP (−)/PGII (+)	34/213	13.77	6.96 (4.70–10.31)	**<0.001**	3.58 (2.29–5.57)	**<0.001**
HP (+)/PGII (+)	348/1969	15.02	7.71 (6.38–9.32)	**<0.001**	6.93 (5.64–8.50)	**<0.001**
HP (−)/PGR (−)	146/7448	1.92	1		1	
HP (+)/PGR (−)	307/5190	5.58	3.02 (2.47–3.69)	**<0.001**	2.96 (2.40–3.65)	**<0.001**
HP (−)/PGR (+)	60/267	18.35	12.12 (9.97–14.72)	**<0.001**	6.40 (4.42–9.28)	**<0.001**
HP (+)/PGR (+)	432/1819	19.19	11.46 (8.29–15.86)	**<0.001**	9.66 (7.84–11.90)	**<0.001**

***** Adjusted for sex, age, body mass index, family history of GC, smoking, and alcohol and salt intake. Values that are statistically significant, with a *p*-value of < 0.05, are displayed in bold. n, number of cases; N, number of controls; GC, gastric cancer; OR, odds ratio; CI, confidence interval; aOR, adjusted odds ratio; HP, *Helicobacter pylori*; HP(−), HP status negative; HP(+), HP status positive; PGII (−), serum pepsinogen II < 21 ng/mL; PGII (+), serum pepsinogen II ≥ 21 ng/mL; PGR (−), pepsinogen ratio > 3; PGR (+), pepsinogen ratio ≤ 3.

## Data Availability

The datasets generated and/or analyzed during the current study are available from the corresponding author upon reasonable request.

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
