# Peer review of "Assessing Serum Pepsinogen and Helicobacter pylori Tests for Detecting Diffuse-Type Gastric Cancer: Insights from a Large-Scale and Propensity-Score-Matched Study in Republic of Korea"

_cancers, 2025, doi:10.3390/cancers17060955_

Round 1

Reviewer 1 Report

Comments and Suggestions for Authors

Dear authors,

Congratulation on a well designed and well written research paper with large sample size.

Introduction and Discussion are well written.

However there are several minor corrections that the authors need to improve before this manuscript can be accepted for publication especially in Results and Materials and Methods sections.

All tables are very hard to read especially Table 1, 2, and 3. The authors should make them more readable so the readers can grasp information immediately from the tables alone. Perhaps the authors can simplify the table to include only essential information and the rest of data can be put into supplementary materials.

Line 141: The authors mentioned that they used latex-enhanced turbidimetric 141 immunoassay to detect serum level of PGI and PGII. Perhaps the authors can include previous study that has used this kit to measure the sera because there is no mention on sensitivity and specificity of this kit. Is there any validation performed by using other tests to ensure that the sera measured by this kit is valid?

Author Response

  1. All tables are very hard to read especially Table 1, 2, and 3. The authors should make them more readable so the readers can grasp information immediately from the tables alone. Perhaps the authors can simplify the table to include only essential information and the rest of data can be put into supplementary materials.

Reply: Thank you for your valuable comment. We have modified Tables 1, 2, and 3 to make them simpler, and we have also created supplemental tables (Tables S1 and S3) for the additional data.

  1. Line 141: The authors mentioned that they used latex-enhanced turbidimetric immunoassay to detect serum level of PGI and PGII. Perhaps the authors can include previous study that has used this kit to measure the sera because there is no mention on sensitivity and specificity of this kit. Is there any validation performed by using other tests to ensure that the sera measured by this kit is valid?

Reply: Thank you for the kind comment. We have published at least seven papers [PMID: 18321304, PMID: 19293719, PMID: 20306133, PMID: 20981206, PMID: 25337572, PMID: 30303591, PMID: 32090375] regarding this latex-enhanced turbidimetric immunoassay for detecting serum levels of PGI and PGII. To share our experience, we would like to introduce one paper entitled as “Correlations among endoscopic, histologic and serologic diagnoses for the assessment of atrophic gastritis. J Cancer Prev. 2014, 19, 47–55 (PMID: 25337572)”. Latex-enhanced turbidimetric immunoassay was demonstrated to be more effective for evaluating gastric atrophy compared to histology and endoscopic atrophic grading. The serum PG I/II ratio using this kit showed a significant decrease when the extent of atrophy increased (R2 = 0.837, P<0.001), with a cut-off value of 3.2 to differentiate between the presence and absence of endoscopic atrophic gastritis. The serum PG I and PG I/II ratios significantly reduced as histological atrophic gastritis progressed, with a cut-off value of 3.0 for a diagnosing histological atrophic gastritis. There was a significant correlation between endoscopic and histological atrophic gastritis, with the sensitivity and specificity of endoscopic diagnosis being 65.9% and 58.0% for antrum and 71.3% and 53.7% for corpus, respectively. Taken together the endoscopic, histological, and serological atrophic gastritis showed relatively good correlations [PMID: 25337572]. The sensitivity and specificity of a PGR of ≤ 3.0 for the detection of dysplasia or cancer were 55.8–62.3% and 61%, respectively [PMID: 18321304].

We have added information about the kit to Methods section and the relevant references as below.

Page 5, Line 161:

The latex-enhanced turbidimetric immunoassay kit demonstrated that serum PGI and PGR levels were significantly lower when histological atrophic gastritis progressed, with a cut-off value was 3.0 for a diagnosing histological atrophic gastritis. A significant correlation was noted between endoscopic and histological atrophic gastritis, with the sensitivity and specificity of endoscopic diagnosis being 65.9% and 58.0% for the antrum, and 71.3% and 53.7% for corpus, respectively [42]. The sensitivity and specificity of a PGR of ≤ 3.0 for detecting dysplasia or cancer were 55.8–62.3% and 61%, respectively [30]. In addition, this kit was also validated in previous studies [43–45]

Page 18, Line 646:

  1. Lee, J.Y.; Kim, N.; Lee, H.S.; Oh, J.C.; Kwon, Y.H.; Choi, Y.J.; Yoon, K.C.; Hwang, J.J.; Lee, H.J.; Lee, A.; et al. Correlations among endoscopic, histologic and serologic diagnoses for the assessment of atrophic gastritis. J. Cancer Prev. 2014, 19, 47–55.
  2. 43. Kim, H.Y.; Kim, N.; Kang, J.M.; Park, Y.S.; Lee, D.H.; Kim, Y.R.; Kim, J.S.; Jung, H.C.; Song, I.S. Clinical meaning of pepsinogen test and Helicobacter pylori serology in the health check-up population in Korea. J. Gastroenterol. Hepatol. 2009, 21, 606–612. 
  3. Yun, C.Y.; Kim, N.; Lee, J.; Lee, J.Y.; Hwang, Y.J.; Lee, H.S.; Yoon, H.; Shin, C.M.; Park, Y.S.; Kim, J.W.; et al. Usefulness of OLGA and OLGIM system not only for intestinal type but also for diffuse type of gastric cancer, and no interaction among the gastric cancer risk factors. Helicobacter 2018, 23, e12542.  
  4. Noh, G.; Kim, N.; Choi, Y.; Lee, H.S.; Hwang, Y.J.; Kim, H.J.; Yoon, H.; Shin, C.M.; Park, Y.S.; Lee, D.H. Long-term follow up of serum pepsinogens in patients with gastric cancer or dysplasia after Helicobacter pylori eradication. J. Gastroenterol. Hepatol. 2020, 35, 1540–1548.

Reviewer 2 Report

Comments and Suggestions for Authors

I really enjoyed reading this manuscript, particularly because of the possibility that serum pepsinogen could be used as an early biomarker of gastric cancer. The authors conclude that a form of serum pepsinogen, PGII level combined with HP status, can be used as a screening test for identifying high-risk GC subjects.

- Here are two main concerns: 1) how would you diagnose HP infection? Would you recommend UBT or serum HPIgG to avoid the more invasive test, EGD? Have the authors evaluated the sensitivities and specificities of rapid HP detection tests? Furthermore, serology for HP detecting IgG could be indicative of a previous and not a current infection. 2) an early biomarker for gastric cancer, as shown by the authors, relies on both high PGII levels and HP infection. Therefore, ensuring that HP positivity is accurately diagnosed is very important.

- Table 4 needs to be modified, or the editorial office can do this. The line numbers are included in the Table's left column.

- Please read through for spelling and typos. Make sure H. pylori is italicized.

- Can a limitation of the study be that patients who underwent HP eradication were not excluded from the study?

Author Response

  1. Here are two main concerns:

1) how would you diagnose HP infection? Would you recommend UBT or serum HPIgG to avoid the more invasive test, EGD? Have the authors evaluated the sensitivities and specificities of rapid HP detection tests? Furthermore, serology for HP detecting IgG could be indicative of a previous and not a current infection.

Reply: Thank you for your valuable comment. Patients were considered to be infected with HP if at least one of four test results (histology, HP polymerase chain reaction, rapid urease test, or [13C]-urea breath test) yielded positive results.

We evaluated the sensitivities and specificities of the rapid urease test and urea breath test. Shin et al. reported that for the overall study population (n = 651), the culture had a specificity of 100% with a low sensitivity (56.2%; 51.1–60.6%), whereas the rapid urease test had a high specificity (96.7%; 92.0–98.8%), but relatively low sensitivity (80.4%; 76.6–83.7%) [PMID: 19889068]. The urea breath test (UBT) using 13C-labeled urea has been reported to have a sensitivity and specificity of > 90% in detecting Helicobacter pylori infection [PMID: 7540995 and PMID: 15569102], and our group reported that its sensitivity and specificity for a cutoff value of 2.5‰ were 99.3% and 47.1%, respectively [PMID: 25640474].

We have added the information about the UBT to the Methods section for readers and the relevant references as below.

Page 5, Line 173:

Our group has reported that the rapid urease test had 96.7% specificity and 80.4% sensitivity [46] and that urea breath test’s sensitivity and specificity for a cutoff value of 2.5‰ were 99.3% and 47.1%, respectively [47].

Page 18, Line 656:

  1. Shin, C.M.; Kim, N.; Lee, H.S.; Lee, H.E.; Lee, S.H.; Park, Y.S.; Hwang, J.H.; Kim, J.W.; Jeong, S.H.; Lee, D.H.; et al. Validation of diagnostic tests for Helicobacter pylori with regard to grade of atrophic gastritis and/or intestinal metaplasia. Helicobacter 2009, 14, 512–529.
  2. Kwon, Y.H.; Kim, N.; Lee, J.Y.; Choi, Y.J.; Yoon, K.; Hwang, J.J.; Lee, H.J.; Lee, A.; Jeong, Y.S.; Oh, S.; et al. The diagnostic validity of citric acid-free, high dose (13)C-urea breath test after Helicobacter pylori eradication in Korea. Helicobacter 2015, 20, 159–168.

We agree that serology for HP detecting IgG could indicate of a previous and not a current infection. However, we included the participants who tested positive for HPIgG as HP status-positive because of an undefined “point of no return,” as described in the text (line 180-181). However, this point you raised may be a limitation of our study. Therefore, we added the following to the limitations in the Discussion section.

Page 14, Line 462:

First, a positive HP status did not exclusively represent active infections. In this study, tests reflecting active HP infection, such as the rapid urease test, urea breath test, culture, HP polymerase chain reaction, or histology, were not performed on all participants. In cases where results from such active tests were unavailable, the serological test results (HPIgG) were used. Those with a history of previous HP eradication were also included. It was recognized that a positive HPIgG result or a history of HP eradication indicates a past HP infection and considered that a past infection could develop into a precancerous lesion.

2) an early biomarker for gastric cancer, as shown by the authors, relies on both high PGII levels and HP infection. Therefore, ensuring that HP positivity is accurately diagnosed is very important.

Reply: We agree that it is crucial to diagnose HP positives accurately. However, if screening is needed in a large population, serology may be preferable because even a past infection may still have the potential to progress to a premalignant condition.

  1. Table 4 needs to be modified, or the editorial office can do this. The line numbers are included in the Table's left column.

Reply: Thank you for kind comment. We tried deleting the existing Table 4 and pasting it again, which seemed to have solved the problem.  

  1. Please read through for spelling and typos. Make sure H. pyloriis italicized.

Reply: Thank you for your attention to detail. We thoroughly checked the text for typographical and other errors and corrected them, including those given below.

Page 1, Line 32: anti-HP antibodies → anti-HP antibody

Page 3, Line 93: and weaker association → and a weaker association

Page 3, Line 94: generally negative → generally lacking

Page 5, Line 134: Helicobacter pylori → Helicobacter pylori

Page 5, Line 171: the Campylobacter-like organism test, → rapid urease test,

Page 5, Line 171: (addition) or [13C]-urea breath test,

Page 5, Line 172: Positive results for → A positive result for

Page 5, Line 172: three tests → four tests

Page 5, Line 176: three other HP tests → other four HP tests

Page 6, Line 192: “moderate/severe” → “severe”

Page 7, Line 248: Table 1 → Table S2

Page 9, Line 267: lower than those in → lower than that in

Page 9, Line 274: (Table S1) → (Table S3)

Page 12, Line 327: (Table S2) → (Table S4)

Page 12, Line 338: all p < 0.001 → all P < 0.001

Page 12, Line 350: (Table S3–S6) → (Table S5–S8)

Page 12, Line 352: (Table S3) → (Table S5)

Page 12, Line 353: (Table S4) → (Table S6)

Page 12, Line 355: (Table S5) → (Table S7)

Page 12, Line 355: (Table S6) → (Table S8)

Page 14, Line 447: 6 GC cases were missing → 6 GC cases were missed

Page 14, Line 448: (Table S7) → (Table S9)

Page 15, Line 469: First → Second

Page 15, Line 474: Second → Third

  1. Can a limitation of the study be that patients who underwent HP eradication were not excluded from the study?

Reply: Since we believe that the possibility of progression to gastric tumors cannot be excluded in patients with past infection with HP (e.g., HP antibody positivity and previous HP eradication), we included the participants with a history of previous HP eradication. However, it is impossible to accurately reflect current HP infections. Therefore, we added the following section to the limitation in Discussion section. This correction is a duplicate of #1.

Page 14, Line 462:

First, a positive HP status did not exclusively represent active infections. In this study, tests reflecting active HP infection, such as the rapid urease test, urea breath test, culture, HP polymerase chain reaction, or histology, were not performed on all participants. In cases where results from such active tests were unavailable, the serological test results (HPIgG) were used. Those with a history of HP eradication were also included. It was recognized that a positive HPIgG result or a history of HP eradication indicates a past HP infection and considered that a past infection could develop into a precancerous lesion.

Reviewer 3 Report

Comments and Suggestions for Authors

Gastric cancer represents annoying health problem in South East Asia, The early detection represents the corner-stone in prevention and treatment of such disease with high incidence rate. The authors investigate the association of serum PGs, including PGII, with GC occurrence, after adjusting for other GC risk factors, using large-scale health checkup data. Also, they determine the risk of GC occurrence according to the combination of each PG value and HP status, particularly focusing on DGC. This is a very important clinical-valued study as they aim to reach to a serologic markers for early detection of DGC for general population screening.

The authors have a good design for their manuscript, have a large number of included patients, exclude the effect of the confounder factors, apply a good statistical analysis that leads to valuable results. The discussion was a comparehensive and  compared their results with the previous studies from different groups in a critical way.

The inclusion of the HP infection status is of much value as this is a WHO-classified carcinogen infection. The role of HP infection in the DGC incidence in South East Asia should be determined. All these aspects had been included and discussed by the authors.

The conclusion they reached can be clinically applied with a good predictive value as a screening tool on wide scale. 

Author Response

Gastric cancer represents annoying health problem in South East Asia, The early detection represents the corner-stone in prevention and treatment of such disease with high incidence rate. The authors investigate the association of serum PGs, including PGII, with GC occurrence, after adjusting for other GC risk factors, using large-scale health checkup data. Also, they determine the risk of GC occurrence according to the combination of each PG value and HP status, particularly focusing on DGC. This is a very important clinical-valued study as they aim to reach to a serologic markers for early detection of DGC for general population screening.

The authors have a good design for their manuscript, have a large number of included patients, exclude the effect of the confounder factors, apply a good statistical analysis that leads to valuable results. The discussion was a comparehensive and compared their results with the previous studies from different groups in a critical way.

The inclusion of the HP infection status is of much value as this is a WHO-classified carcinogen infection. The role of HP infection in the DGC incidence in South East Asia should be determined. All these aspects had been included and discussed by the authors.

The conclusion they reached can be clinically applied with a good predictive value as a screening tool on wide scale.

Reply: Thank you for your positive evaluation. We did our best to respond to the detailed comments of the other three reviewers.

Reviewer 4 Report

Comments and Suggestions for Authors

Manuscript ID: cancers-3474484 review 1

Title: Assessing serum pepsinogen and Helicobacter pylori tests for detecting diffuse-type gastric cancer: Insights from a large-scale and propensity score matched study in South Korea

Authors: Lim et al.

The manuscript is interesting and generally well written. The manuscript presents the large-scale (N = 23,015) case-control study conducted to validate the utility of pepsinogen in diagnosing gastric cancer during health checkups. Authors found that pepsinogen II (PGII)-positive or PGR (PGI-to-PGII ratio)-positive, but not PGIR(reciprocal PGR or PGII-to-PGI ratio)-positive status, were independent predictors of gastric cancer the general population, after PSM (propensity-score matching). The authors stated that pepsinogen II showed a greater association with diffuse-type and the pepsinogen ratio showed a greater association with intestinal-type gastric cancer as well as diffuse-type gastric cancer. The combination of high pepsinogen II and positive H. pylori status was associated with a heightened risk of early-stage diffuse-type gastric cancer, especially in young female participants. The studies presented are consistent with current research in this field. The results are well documented. It should also be noted that there were minor imperfections during the preparation of the manuscript that should be corrected before publication.

List of some deficiencies

  1. All abbreviations should be presented in their full name at the point where they appear for the first time, starting from the abstract. Full names of abbreviation should be repeated in each main section of the manuscript, as well as in each Figure legends and Tables. Figures and their legends must be understandable without references to the body of the manuscript. On the other hand, the number of abbreviations should be reduced. The excess of abbreviations makes the manuscript difficult to understand. Adding a list of abbreviations at the end of the manuscript is a good idea and helps in emergency situations. However, reading the manuscript should be easy for the reader and possible without having to constantly return to the list of abbreviations. For examples some irregularities: abstract, line 30, the authors used the abbreviation “PGII” without giving its meaning, they can introduce this abbreviation in line 22; abstract, line 36, before “PGII ≥21 ng/mL”, the authors should add “serum”; Introduction, line 77, “PGI:II ratio” replace with “PGI-to-PGII ratio”.
  2. Introduction, after two first sentences, the authors should state “The crude global prevalence of Helicobacter pylori has reduced from 52.6% before 1990 to 43.9% in adults during 2015 through 2022, but was as still as high as 35.1% in children and adolescents during 2015 through 2022. Significant reduction of Helicobacter pylori was observed in adults in the Western Pacific, Southeast Asian, and African regions. However, H pylori prevalence was not significantly reduced in children and adolescents in any World Health Organization regions. The incidence of gastric cancer has decreased globally and in various countries where the prevalence of H pylori infection has declined (PMID: 38176660)
  3. Introduction, the third sentence “Eastern Asia has the highest GC incidence rate, at 45.7 cases per 100,000 persons, amounting for 60.3% of new GC cases and 56.6% of GC-related deaths worldwide”. It is indeed a huge incidence of stomach cancer. However, it would be advisable for the authors to state whether in recent years there has been an increase or decrease in the incidence of stomach cancer in South Korea.
  4. In the abstract and introduction, the authors stated that the aim of the study was “to validate the role of pepsinogens with or without Helicobacter pylori HP) in gastric cancer (GC) screening, focusing on PGII and diffuse-type GC (DGC), using large-scale health checkup data. On the other hand, endoscopic screening of large populations is effective in countries with a high prevalence of gastric cancer, such as South Korea and Japan. In countries with high prevalence of gastric cancer; endoscopic screening reduced the gastric cancer-related mortality rate by 47% in a nested case-control study (PMID: 28147224) For the above reasons, it would be advisable for the authors to present the real significance of serological tests in screening for gastric cancer in Korea in relation to endoscopic examination. If the result of a serological test indicates a risk of stomach cancer, it will end with an endoscopic examination anyway. However, if the serological test does not indicate a risk of stomach cancer, it is recommended not to perform an endoscopic test? The reviewer found no information in the manuscript about possible false negative serological test results. Did they occur? If so, then abandoning the endoscopic examination would lead to a delay in making the correct diagnosis in such cases, and this would be associated with a threat to the patient's life. This problem should be presented in the Discussion and Conclusions.
  5. Using the keywords "pepsinogen gastric cancer" in PubMed yields 1182 results. The authors' manuscript contains only 42 references. The authors should present more detailed work of their predecessors dealing with the same research topic.

Author Response

List of some deficiencies

1. All abbreviations should be presented in their full name at the point where they appear for the first time, starting from the abstract. Full names of abbreviation should be repeated in each main section of the manuscript, as well as in each Figure legends and Tables. Figures and their legends must be understandable without references to the body of the manuscript. On the other hand, the number of abbreviations should be reduced. The excess of abbreviations makes the manuscript difficult to understand. Adding a list of abbreviations at the end of the manuscript is a good idea and helps in emergency situations. However, reading the manuscript should be easy for the reader and possible without having to constantly return to the list of abbreviations. For examples some irregularities: abstract, line 30, the authors used the abbreviation “PGII” without giving its meaning, they can introduce this abbreviation in line 22; abstract, line 36, before “PGII ≥21 ng/mL”, the authors should add “serum”; Introduction, line 77, “PGI:II ratio” replace with “PGI-to-PGII ratio”.

Reply: Thank you for your valuable feedback. As you recommended, the abbreviations were defined separately in the abstract and main text, and the abbreviations used fewer than four times in the text were removed to enhance readability. We also corrected the all abbreviations used in subheading and titles of tables and figure 2.

A list of corrections is given below.

Page 1, Line 21: pepsinogen II → serum pepsinogen II (PGII)

Page 1, Line 23: pepsinogen II → PGII/ H. pyloriHelicobacter pylori

Page 1, Line 30: PGII → serum pepsinogen II (PGII)

Page 1, Line 36, Line 42, & Line 43: PGII → serum PGII

Page 3, Line 71: Helicobacter pylori (HP)-related → HP-related

Page 3, Line 87: PGI:II ratio → PGI-to-PGII ratio

Page 5, Line 148: Serologic tests for HP Ig G antibody and PG series Serologic tests for Helicobacter pylori antibody and pepsinogen series

Page 5, Line 169: HP infection history assessment Helicobacter pylori status assessment

Page 6, Line 202–203: receiver operating characteristic (ROC) curves → receiver operating characteristic curves

Page 6, Line 207–208: year of birth (YOB) → year of birth

Page 6, Line 209 & Line 215: YOB → year of birth

Page 7, Line 252: PSMpropensity-score matching

Page 8, legend of revised table 1: (addition) PGI, serum pepsinogen I; PGII, serum pepsinogen II. Subjects with missing data were shown in Table S1.

Salty diet scores: “not/mild” (score 0~2) and “severe” (score 3~4) assessed via two questions [“Do you add extra salt or soy sauce to your food when you eat?” and “Do you eat salty foods such as salted seafood, pickled vegetables, and soup-based meals?” which had three kinds of answers [“never (score 0), sometimes (score 1), frequently (score 2)”], in the questionnaire.

Page 8, Line 253: PSMpropensity-score matching

Page 9, legend of revised table 2: (addition) PGI, serum pepsinogen I; PGII, serum pepsinogen II. Subjects with missing data were shown in Table S1.

Salty diet scores: “not/mild” (score 0~2) and “severe” (score 3~4) assessed via two questions [“Do you add extra salt or soy sauce to your food when you eat?” and “Do you eat salty foods such as salted seafood, pickled vegetables, and soup-based meals?” which had three kinds of answers [“never (score 0), sometimes (score 1), frequently (score 2)”], in the questionnaire.

Page 9, Line 268: Association of each PG value or HP status with gastric neoplasm after propensity-score matching Association of each pepsinogen value or Helicobacter pylori status with gastric neoplasm after propensity-score matching

Page 9, Line 282: PSMpropensity-score matching

Page 10, Line 285-287: PSM, propensity score matching; PGII pos (PGII- positive), PGI I≥ 21ng/ml; PGII neg (PGII- negative), PGII < 21ng/ml; PGIR pos (PGIR-positive), PGI ≤ 70 and PGR ≤ 3; PGIR neg (PGIR-negative), PGI > 70 or PGR > 3; PGR pos (PGR-positive), PGI/II i.e. PGR ≤ 3; PGR neg (PGR-negative), PGR > 3; n, number of cases; N, number of controls;

→ PGII neg (PGII- negative), serum pepsinogen II < 21ng/ml; PGII pos (PGII- positive), serum pepsinogen II ≥ 21ng/ml; PGIR neg (PGIR-negative), serum pepsinogen I > 70 or pepsinogen ratio > 3; PGIR pos (PGIR-positive), serum pepsinogen I ≤ 70 and pepsinogen ratio ≤ 3; PGR neg (PGR-negative), pepsinogen ratio > 3; PGR pos (PGR-positive), pepsinogen ratio ≤ 3;

Page 10, Line 302: Detection power of PG values for diagnosis of GC Detection power of pepsinogen values for diagnosis of gastric cancer

Page 11, Line 310: title of figure 2: corresponding AUC of each PG valuecorresponding area under the curve of each pepsinogen value

Page 11, Line 311: legend of figure 2: The AUC for PGII → The AUC for serum pepsinogen II (PGII)

Page 11, Line 313: legend of figure 2: The AUC for 1/PGR → The AUC for reciprocal pepsinogen ratio (1/PGR)

Page 11, Line 316: legend of figure 2: PGII, pepsinogen II ≥ 21 ng/mL → PGII, serum pepsinogen II ≥ 21 ng/mL  

Page 11, Line 318: legend of figure 2: (addition) 1/PGR, reciprocal pepsinogen ratio

Page 12, Line 328: combining PG values and HP status in GC combining pepsinogen values and Helicobacter pylori status in gastric cancer

Page 12, Line 341: PSMpropensity-score matching

Page 12, Line 345-346: legend of table 4: PGII (-), PGII < 21 ng/ml; PGII (+), PGII ≥ 21 ng/ml; PGR (-), PGI/II ratio > 3; PGR (+), PGI/II ratio ≤ 3

→ PGII (-), serum pepsinogen II < 21 ng/ml; PGII (+), serum pepsinogen II ≥ 21 ng/ml; PGR (-), pepsinogen ratio > 3; PGR (+), pepsinogen ratio ≤ 3

Page 16, Abbreviations: (removed) ROC, receiver operating characteristic; YOB, year of birth

2. Introduction, after two first sentences, the authors should state “The crude global prevalence of Helicobacter pylori has reduced from 52.6% before 1990 to 43.9% in adults during 2015 through 2022, but was as still as high as 35.1% in children and adolescents during 2015 through 2022. Significant reduction of Helicobacter pylori was observed in adults in the Western Pacific, Southeast Asian, and African regions. However, H pylori prevalence was not significantly reduced in children and adolescents in any World Health Organization regions. The incidence of gastric cancer has decreased globally and in various countries where the prevalence of H pylori infection has declined (PMID: 38176660)

Reply: Thank you for highlighting the recent noteworthy results on Helicobacter pylori prevalence and gastric cancer incidence. As you recommended, we added relevant content in the Introduction section.

Page 3, Line 51:

According to a recent meta-analysis, the crude global prevalence of Helicobacter pylori (HP) has reduced from 52.6% before 1990 to 43.9% from 2015 to 2022 in adults, and the incidence of GC has decreased globally and in various countries where the prevalence of HP infection has declined [2].

Page 16, Line 551:

  1. Chen, Y.C.; Malfertheiner, P.; Yu, H.T.; Kuo, C.L.; Chang, Y.Y.; Meng, F.T.; Wu, Y.X.; Hsiao, J.L.; Chen, M.J.; Lin, K.P.; et al. Global prevalence of Helicobacter pylori infection and incidence of gastric cancer between 1980 and 2022. Gastroenterology 2024, 166, 605–619.

3. Introduction, the third sentence “Eastern Asia has the highest GC incidence rate, at 45.7 cases per 100,000 persons, amounting for 60.3% of new GC cases and 56.6% of GC-related deaths worldwide”. It is indeed a huge incidence of stomach cancer. However, it would be advisable for the authors to state whether in recent years there has been an increase or decrease in the incidence of stomach cancer in South Korea.

Reply: Thank you for the important comment. As you pointed out, the incidence of gastric cancer in East Asia, including South Korea, Japan, and China, has been declining in recent years. However, according to recent reports, all three countries still exhibit an age-standardized incidence rate of approximately 30 per 100,000 individuals, which remains higher than global levels, highlighting the need for continued efforts to sustain this downward trend. In consideration of your advice, we have incorporated the following additions into the Introduction section.

Page 3, Line 57:

Although the incidence and mortality of GC have been declining owing to the increased diagnosis of precursor lesions and early GC following the introduction of nationwide screening programs [2-4], as well as advancements in treatment techniques, the age-standardized incidence rates of GC in 2019 were still higher than those in other countries, at 28.67, 28.29, and 30.64 per 100,000 individuals in Korea, Japan, and China, respectively [6].

Page 16, Line 561:

  1. Yang, X.; Zhang, T.; Zhang, H.; Sang, S.; Chen, H.; Zuo, X. Temporal trend of gastric cancer burden along with its risk factors in China from 1990 to 2019, and projections until 2030: comparison with Japan, South Korea, and Mongolia. Biomark. Res. 2021, 9, 84.

4. In the abstract and introduction, the authors stated that the aim of the study was “to validate the role of pepsinogens with or without Helicobacter pylori HP) in gastric cancer (GC) screening, focusing on PGII and diffuse-type GC (DGC), using large-scale health checkup data. On the other hand, endoscopic screening of large populations is effective in countries with a high prevalence of gastric cancer, such as South Korea and Japan. In countries with high prevalence of gastric cancer; endoscopic screening reduced the gastric cancer-related mortality rate by 47% in a nested case-control study (PMID: 28147224) For the above reasons, it would be advisable for the authors to present the real significance of serological tests in screening for gastric cancer in Korea in relation to endoscopic examination. If the result of a serological test indicates a risk of stomach cancer, it will end with an endoscopic examination anyway. However, if the serological test does not indicate a risk of stomach cancer, it is recommended not to perform an endoscopic test? The reviewer found no information in the manuscript about possible false negative serological test results. Did they occur? If so, then abandoning the endoscopic examination would lead to a delay in making the correct diagnosis in such cases, and this would be associated with a threat to the patient's life. This problem should be presented in the Discussion and Conclusions.

Reply: Thank you for your insightful comments. We agree with your point, and we have had the same concerns. In a country like Korea, where access to endoscopy is relatively easy and the incidence of gastric cancer is somewhat high, we are not suggesting that we should screen for gastric cancer at all ages without endoscopy, but we are suggesting that we should screen asymptomatic individuals under the age of 40 who are not eligible for the national gastric cancer screening program, first by serologic testing. Using this approach in the current study, in the group of people under 40 years of age who were serologically negative and did not undergo endoscopy, there were six patients with gastric cancer (line 445-446), and the false negative rate (1-sensitivity) was 9.52%.

We have clarified this in the Discussion section and modified the Table S9 by addition of the column of “false negative rate” in Table S9.

Page 14, line 447:

A total of 6 GC cases were missed due to failure to filter out by serologic results under 40 years (Table S9), and the false negative rate (1-sensitivity) was 9.52%. In other words, if an endoscopy is not performed solely based on serologic results, approximately 9.5% of GC patients may be missed. This is an area that needs to be approached with caution. To compensate for this, the presence or absence of symptoms and other GC risk factors such as a family history of GC or smoking should be sufficiently considered.

Supplementary material, Table S9: (The table won't paste. Please see the attached file)

5. Using the keywords "pepsinogen gastric cancer" in PubMed yields 1182 results. The authors' manuscript contains only 42 references. The authors should present more detailed work of their predecessors dealing with the same research topic.

Reply: Thank you for your insightful comments. Research on utilizing serologic tests, including pepsinogen, for the diagnosis of precursor lesions or early gastric cancer has been ongoing. However, in recent years, there have been relatively few newly reported studies on this topic, as well as large-scale studies applying these methods in clinical practice. Therefore, only a limited number of results were cited in our paper initially. Following your advice, we have reviewed the latest research findings on gastric cancer screening using serologic tests and have supplemented the Introduction section as follows.

Page 3, Line 87:

As low PGI and PGR levels reflect gastric atrophy, these markers have been studied to identify high-risk patients with conditions such as atrophic gastritis or intestinal metaplasia [19,20] or to detect neoplastic lesions at an earlier stage [21]. Mechanistically, these markers are expected to be particularly useful for diagnosing intestinal-type gastric cancer (IGC), which follows Correa’s cascade [16,17].

Page 4, Line 106:

Some studies have highlighted the limitations of diagnostic accuracy when using serologic tests alone [32], and others have suggested that assessing the HP infection status concurrently is necessary for clinical utility [33,34], showing a high diagnostic rate of GC in past or current HP infection groups. Recently, our group reported that PGII ≥ 20 ng/mL and a positive HP status may help detect early-stage DGC (DGC-E) in a young female group in a tertiary hospital, although not in the general population [35]. In addition, a recent study has reported the relationship between HP infection and progression of DGC [36].

Page 17, Line 587:

  1. Zhou, X.; Zhu, H.; Zhu, C.; Lin, K.; Cai, Q.; Li, Z.; Du, Y. Helicobacter pylori infection and serum pepsinogen level with the risk of gastric precancerous conditions: a cross-sectional study of high-risk gastric cancer population in China. Clin. Gastroenterol. 2021, 55, 778–784.
  2. Sánchez-López, J.Y.; Díaz-Herrera, L.C.; Rizo-de la Torre, L.D.C. Pepsinogen I, pepsinogen II, gastrin-17, and Helicobacter pylori serological biomarkers in the diagnosis of precursor lesions of gastric cancer. Med. Sci. 2024, 20, 1016–1021.
  3. In, H.; Sarkar, S.; Ward, J.; Friedmann, P.; Parides, M.; Yang, J.; Epplein, M. Serum pepsinogen as a biomarker for gastric cancer in the United States: a nested case-control study using the PLCO cancer screening trial data. Cancer Epidemiol. Biomarkers Prev. 2022, 31, 1426–1432.

Page 18, Line 618:

32. Gašenko, E.; Bogdanova, I.; Sjomina, O.; Aleksandraviča, I.; Kiršners, A.; Ancāns, G.; Rudzīte, D.; Vangravs, R.; Sīviņš, A.; Škapars, R.; et al. Assessing the utility of pepsinogens and gastrin-17 in gastric cancer detection. J. Cancer Prev. 2023, 32, 478–484.

33. Hatta, ; Koike, T.; Asonuma, S.; Okata, H.; Uno, K.; Oikawa, T.; Iwai, W.; Yonechi, M.; Fukushi, D.; Kayaba, S.; et al. Smoking history and severe atrophic gastritis assessed by pepsinogen are risk factors for the prevalence of synchronous gastric cancers in patients with gastric endoscopic submucosal dissection: a multicenter prospective cohort study. J. Gastroenterol. 2023, 58, 433–443.

34. Hirai, R; Hirai, M.; Otsuka, M.; Mitsuhashi, T.; Shimodate, Y.; Mouri, H.; Matsueda, K.; Yamamoto, H.; Mizunom M. Endoscopic evaluation by the Kyoto classification of gastritis combined with serum anti-Helicobacter pylori antibody testing reliably risk-stratifies subjects in a population-based gastric cancer screening program. Gastroenterol. 2023, 58, 848–855.

36. Kang, S.; Park, M.; Cho, J.Y.; Ahn, S.J.; Yoon, C.; Kim, S.G.; Cho, S.J. Tumorigenic mechanisms of estrogen and Helicobacter pylori cytotoxin-associated gene A in estrogen receptor alpha-positive diffuse-type gastric adenocarcinoma. Gastric Cancer 2022, 25, 678–696.

Reviewer 5 Report

Comments and Suggestions for Authors

The manuscript (including table and figure) is so poorly summarized that it is hard to understand the contents. For example, table 1 as well as table 2 are too large and too complicated to follow.  Classification of gastric lesions, from adenoma to advanced diffuse gastric cancer, is too meticulous. In summary, I am unable to identify what the authors are addressing in the manuscript, though the abstract is well summarized. 

Author Response

Comments and Suggestions for Authors

The manuscript (including table and figure) is so poorly summarized that it is hard to understand the contents. For example, table 1 as well as table 2 are too large and too complicated to follow.  Classification of gastric lesions, from adenoma to advanced diffuse gastric cancer, is too meticulous. In summary, I am unable to identify what the authors are addressing in the manuscript, though the abstract is well summarized.

Reply: Thank you. We did our best to respond to the detailed comments of the other three reviewers.

Round 2

Reviewer 4 Report

Comments and Suggestions for Authors

The manuscript is ready for publication.

Author Response

Comments and Suggestions for Authors: The manuscript is ready for publication.

Reply: Thank you for your positive feedback. We appreciate you letting us make this manuscript more informative.

Reviewer 5 Report

Comments and Suggestions for Authors

Insufficient improvement (contents of tables are still redundant)

Author Response

Comments and Suggestions for Authors: Insufficient improvement (contents of tables are still redundant)

Reply: Thank you for making the manuscript much more neatly. Compared to the last revision, we have tried to make Table 1, 2, and 3 shorter and less redundant, while still including the messages we want to convey. And the parts that were deleted from the original tables but needed for the explanation in the main text were created as supplementary tables (Table S1 and S5).

There were the changes to the main text as the tables were simplified and supplementary tables were added. They were highlighted with green.

Page 7, Line 246: (addition) Table S1

Page 7, Line 251: (addition) and Table S1

Page 8, Table 1 foot note: (addition) Basal characteristics of the subjects with gastric adenoma were shown in Table S1 and

Page 8, Table 1 foot note: Table S1 → Table S3

Page 8, Line 258: (addition) and Table S1

Page 8, Line 260: (addition) and Table S1

Page 9, Table 2 foot note: (addition) Basal characteristics of the subjects with gastric adenoma were shown in Table S1 and

Page 9, Table 2 foot note: Table S1 → Table S3

Page 9, Line 271: (addition) (Table S1)

Page 10, Line 281: Table S3 → Table S4

Page 10, Table 3 footnote: (deletion) IGC-E, intestinal-type gastric cancer early-stage; IGC-A, intestinal-type gastric cancer advanced-stage; DGC-E, diffuse-type gastric cancer early-stage; DGC-A, diffuse-type gastric cancer advanced-stage

Page 10, Line 306: (addition) ; Table S5

Page 10, Line 307: Table 3 → Table S5

Page 12, Line 336: Table S4 → Table S6

Page 13, Line 376: Table S5–S8→ Table S7–S10

Page 13, Line 378: Table S5 → Table S7

Page 13, Line 379: Table S6 → Table S8

Page 13, Line 381: Table S7 → Table S9

Page 13, Line 381: Table S8 → Table S10

Page 14, Line 477: Table S9 → Table S11

Page 15, Line 521: (addition) Basal characteristics of the subjects with gastric adenoma were shown in Table S1 and

Page 15, Line 522: Table S1 → Table S3

Page 15, Line 523: Table S3 → Table S4

Page 15, Line 524: Table S5: (addition) Multivariable-adjusted logistic regression analysis for subtypes of gastric cancers according to pepsinogen values before and after propensity-score matching;

Page 15, Line 526: Table S4 → Table S6

Page 15, Line 527: Table S5 → Table S7

Page 15, Line 528: Table S6 → Table S8

Page 15, Line 530: Table S7 → Table S9

Page 15, Line 531: Table S8 → Table S10

Page 16, Line 533: Table S9 → Table S11

Round 3

Reviewer 5 Report

Comments and Suggestions for Authors

It is still impossible to understand tables 1 and 2, both of which remain too redundant and too complicated.  Unfortunately, I cannot accept the revised manuscript because of little improvement.